# Natural History of Aerosol-Induced Ebola Virus Disease in Rhesus Macaques

**DOI:** 10.3390/v13112297

**Published:** 2021-11-17

**Authors:** Isaac Downs, Joshua C. Johnson, Franco Rossi, David Dyer, David L. Saunders, Nancy A. Twenhafel, Heather L. Esham, William D. Pratt, John Trefry, Elizabeth Zumbrun, Paul R. Facemire, Sara C. Johnston, Erin L. Tompkins, Nathan K. Jansen, Anna Honko, Anthony P. Cardile

**Affiliations:** 1US Army Medical Research Institute of Infectious Diseases (USAMRIID), Fort Detrick, Frederick, MD 21702, USA; Joshua.Johnson@modernatx.com (J.C.J.); franco.d.rossi.ctr@mail.mil (F.R.); david.n.dyer3.civ@mail.mil (D.D.); david.l.saunders.mil@mail.mil (D.L.S.); nancy.a.twenhafel.mil@mail.mil (N.A.T.); heather.l.esham.civ@mail.mil (H.L.E.); hoomnong@verizon.netl (W.D.P.); john.c.trefry.civ@mail.mil (J.T.); elizabeth.e.zumbrun.civ@mail.mil (E.Z.); paul.r.facemire2.mil@mail.mil (P.R.F.); sara.c.johnston2.civ@mail.mil (S.C.J.); erin.l.tompkins.mil@mail.mil (E.L.T.); nathan.k.jansen.mil@mail.mil (N.K.J.); honko@bu.edu (A.H.); anthony.p.cardile.civ@mail.mil (A.P.C.); 2Moderna, Inc., Cambridge, MA 02139, USA; 3Defense Threat Reduction Agency, Fort Belvoir, VA 22060, USA; 4Investigator at National Emerging Infectious Diseases Laboratories, Boston University School of Medicine, Boston, MA 02118, USA

**Keywords:** Ebola, Kikwit, Zaire, virus, viral hemorrhagic fever, natural history, telemetry, aerosol, respiratory alkalosis, cytokine storm, rhesus macaque, *Macaca mulatta*

## Abstract

Ebola virus disease (EVD) is a serious global health concern because case fatality rates are approximately 50% due to recent widespread outbreaks in Africa. Well-defined nonhuman primate (NHP) models for different routes of Ebola virus exposure are needed to test the efficacy of candidate countermeasures. In this natural history study, four rhesus macaques were challenged via aerosol with a target titer of 1000 plaque-forming units per milliliter of Ebola virus. The course of disease was split into the following stages for descriptive purposes: subclinical, clinical, and decompensated. During the subclinical stage, high levels of venous partial pressure of carbon dioxide led to respiratory acidemia in three of four of the NHPs, and all developed lymphopenia. During the clinical stage, all animals had fever, viremia, and respiratory alkalosis. The decompensatory stage involved coagulopathy, cytokine storm, and liver and renal injury. These events were followed by hypotension, elevated lactate, metabolic acidemia, shock and mortality similar to historic intramuscular challenge studies. Viral loads in the lungs of aerosol-exposed animals were not distinctly different compared to previous intramuscularly challenged studies. Differences in the aerosol model, compared to intramuscular model, include an extended subclinical stage, shortened clinical stage, and general decompensated stage. Therefore, the shortened timeframe for clinical detection of the aerosol-induced disease can impair timely therapeutic administration. In summary, this nonhuman primate model of aerosol-induced EVD characterizes early disease markers and additional details to enable countermeasure development.

## 1. Introduction

Ebolaviruses are lipid enveloped, single-stranded, negative-sense RNA viruses [1,2,3,4]. These filamentous virions are members of the family *Filoviridae*, with a uniform diameter of 80 nm and a length greater than 1000 nm, and are known for causing severe human disease [3,5,6]. *Zaire ebolavirus* is one of six species from the *Ebolavirus* genus; the others are *Bundibugyo ebolavirus*, *Sudan ebolavirus*, *Tai Forest ebolavirus*, *Reston ebolavirus*, and *Bombali ebolavirus* [1,2,7,8]. Ebola virus (EBOV) is classified as a CDC Category A biological agent and Risk Group 4 pathogen capable of aerosol transmission, can spread from person-to-person, and has a high mortality rate [9,10,11,12,13]. 

The largest EVD outbreak began within Western Africa in 2013 and was not contained until 2016. The total number of infected human cases was 28,652, with case fatalities of greater than 11,000, reaching pandemic status as it spread to several distant countries including Spain and the United States [5,14,15]. The second largest EVD outbreak occurred in 2018–2020 in the Democratic Republic of Congo, with at least 3460 total cases and 2280 case fatalities [5,16,17,18]. In response to such outbreaks, the FDA recently approved the use of Inmazeb for the treatment of Ebola-infected pediatric and adult subjects [19]. 

In comparison to human cases, intramuscular EBOV exposure studies of nonhuman primates have shown a high degree of clinical similarity. NHPs exhibit fever, rash, dehydration, systemic inflammation, coagulopathy, lymphoid tissue necrosis with marked lymphopenia, hepatic necrosis, and renal dysfunction [9,20,21,22,23]. Though the natural reservoir has yet to be identified, wild caught fruit bats have been found to express serum antiviral IgG antibodies and to have EBOV genetic material within liver and spleen tissue [24,25,26]. Direct physical contact and exposure to bodily fluids are the primary modes of EBOV transmission. Additional modes of transmission include contact-related funeral practices, shared meals, bush meat consumption, and direct contact with apes and bats [27,28,29]. 

The aerosol transmission of EBOV in humans has not been well-characterized by epidemiological data [12,14,21,26,27,30,31]. Such aerosol transmission could occur during the forceful emission of bodily fluids such as coughing, vomiting, diarrhea, and aerosol-generating medical procedures [12,26,32,33]. Through potential aerosol transmission, Jaax et al. 1995 reported the unintentional EBOV infection of rhesus macaques three meters apart in the same biocontainment facility [33].

The development of relevant EVD models is crucial for understanding disease progression to enable countermeasure assessment: a significant step for protection against the biological threat. The FDA has stated that “natural history studies are studies in which animals are exposed to a challenge agent and monitored to gain an understanding of the development and progression of the resulting disease or condition, … the time from exposure to manifestation onset, time course and order of manifestation progression, and severity” [34]. This study of aerosol-induced EBOV infection in rhesus macaques monitored physiological parameters via telemetry, clinical observations, blood chemistry, complete blood counts, and coagulation factors over the course of the disease. The pathological assessment of this current work was published in the serial sacrifice study to substantiate the endpoint findings [35]. Research findings from both studies will be used to understand the progression of aerosol-induced EVD and later compared to historic intramuscular studies. 

## 2. Materials and Methods

### 2.1. Agent Preparation and Challenge Conditions

The EBOV challenge agent (Ebola virus/H.sapiens-tc/COD/1995/Kikwit-9510621 was derived from the 1995 Kikwit outbreak as previously described (PMID: 23001720) [23,35,36,37,38]. The morphology, nucleotide sequence, purity, sterility, and virulence of the stock (16502; Genbank accession number—JQ352763.1 [39]) were confirmed at USAMRIID prior to use as a challenge agent [23,40,41]. On the day of challenge, the viral stock was serially diluted to a 1:100 final solution with a target titer of 1000 plaque-forming units per milliliter (PFU/mL) using α-MEM-containing GlutaMax (Gibco, Life Technologies, Carlsbad, CA, USA) supplemented with 2% fetal bovine serum. 

All nonhuman primate (NHP) aerosol exposures were performed in a class III biosafety cabinet inside a BSL-4 laboratory using a 3-jet collision nebulizer (BGI Inc., Waltham, MA, USA) and the head-only automated bioaerosol exposure II (ABES-II) system developed at USAMRIID [35,42,43]. NHPs were anesthetized with 3 mg/kg of telazol (Fort Dodge Animal Health, New York, NY, USA) via intramuscular injection. The minute volume of each NHP was determined immediately prior to aerosol exposure using a whole body plethysmography chamber (Buxco Research Systems, Wilmington, NC, USA) to calculate the duration of each exposure (see Appendix A).

A sample of the aerosol spray was captured for each NHP using an all glass impinger (AGI). These samples were evaluated with a plaque assay to determine the viable aerosol concentration within the chamber (Appendix A). The delivered dose was calculated for each NHP by multiplying the total volume (*V_t_*) of experimental atmosphere inhaled (*V_t_* = *V_m_* × length of exposure) by the aerosol concentration (*C_e_*) (“delivered dose” p*Ce* × *V_t_*). This equation assumes constant minute volume and constant aerosol concentration over time with complete (100%) respiratory deposition. Aerosol concentration is calculated by: (C_sampler_ × V_sampler)_/(Q_sampler_ × t_sampled_), where C_sampler_ equals the titrated concentration of the sampler, V_sampler_ equals the volume of the collection media in the sampler, Q_sampler_ equals the flow rate through the sampler, and t_sampled_ equals the total time the sample was taken [43]. NHP1–NHP4 received calculated delivered doses of 878, 743, 1595, and 996 PFU, respectively (the target dose was 1000 PFU). The environmental conditions of the head only chamber were generally 26 °C with 55% relative humidity.

### 2.2. Animals and Telemetry

The Institutional Animal Care and Use Committee at USAMRIID approved the study protocol on 20 March 2009. Research under this approved protocol was performed in accordance with the Public Health Service Policy, Animal Welfare Act, and other federal statutes and regulations relating to animals and experiments involving animals. The primary enclosures complied with the 2011 Guide for the Care and Use of Laboratory Animals and U.S. Department of Agriculture Animal Welfare Act (9 CFR, Parts 1, 2, and 3 [44]. The facility where this research was conducted is accredited by the Association for Assessment and Accreditation of Laboratory Animal Care. Two male and two female rhesus macaques (*Macaca mulatta*) from World Wide Primates, Inc. (Miami, FL, USA), with a median weight of 6.2 kg (range: 5.4–8.5 kg), were chosen for this disease progression study. Pathological assessments of NHPs in this current work were published in Twenhafel et al. 2012 and are marked in parenthesis: NHP1 (NH3F), NHP2 (NH1M), NHP3 (NH4F), and NHP4 (NH2M).

Each NHP was implanted with a T27 Integrated Telemetry Systems radio telemetry device (Konigsberg Instruments, Inc., Pasadena, CA, USA) to gather accurate real-time physiological data. Following the ITS implantation surgery, NHPs recovered for six months prior to Groshong 7F central venous catheter (CVC) implantation (Bard Access Systems, Salt Lake City, UT, USA) for the purpose of blood sample collection from non-anesthetized animals. In order to protect the catheters and allow NHPs the full range of movement within their enclosures, NHPs were fitted with Lomir jackets (Malone, NY, USA) and allowed to recover for at least one week after CVC placement. NHPs were transferred into the BSL-4 environment for one week of acclimatization prior to challenge. Heart rate, temperature, arterial blood pressure, left ventricular pressure, electrocardiogram, and intrathoracic pressure to derive respiratory rate were continuously recorded in each NHP and averaged every thirty minutes throughout the study, as previously described [42]. No mean blood pressure is available for NHP2 because of a nonfunctional pressure sensor.

### 2.3. Daily Observations

All animals were observed daily beginning three days prior to challenge and continuing throughout the study until day post-exposure (DPE) 5. Thereafter, animals were observed twice daily with at least four hours apart starting on DPE 5 and until NHPs met euthanasia criteria or succumbed to the disease. Written observation records for DPE 7 were not found. The evaluated observation parameters included biscuit/fruit consumption, condition of stool, rash, edema, tremors, nasal exudate, urine output, posture, respirations, bleeding, responsiveness, seizure activity, and neurological dysfunction including the ocular, motor, and auditory responses of the animal. Scores were noted for each parameter and ranged from normal (0) to mild (1), moderate (2), and severe (3) except for responsiveness and seizure, which ranged from normal (0) to very mild (1), mild (2), moderate (3), and severe (4). An animal was euthanized under deep anesthesia when the total mean score (sum of changes in responsiveness, posture and appearance, respiration, bleeding, and seizures) of an evaluation was ten or with a seizure score of four. Another animal was also euthanized with a total mean score of eight and noted for potential seizures. 

### 2.4. Blood Sampling

CVCs were flushed with saline from a 10 mL PosiFlush saline syringe (BD Biosciences, San Jose, CA, USA) prior to blood draw. The daily collection of blood samples from each CVC occurred between DPE-3 and DPE 10 with a 2.5 mL volume collected with a 5 mL syringe distributed into 1 mL sodium citrate MiniCollect tubes (Greiner Bio-One, Monroe, NC, USA), 1 mL clot activator tubes (BD Biosciences), and 0.5 mL K2EDTA MAP tubes. Saline was used to again flush the CVC line, which was then maintained by flushing with PosiFlush pre-filled heparin lock syringes (BD Biosciences). These syringes were left attached to the CVC cage-side port until the next sample collection. Animals were anesthetized with ketamine HCl at 10 mg/kg (VedCo, Saint Joseph, MO, USA) or a combination of tiletamine HCl and zolazepam HCl at 3 mg/kg (Fort Dodge Animal Health, New York, NY, USA) on days where sampling through the CVC was complicated due to kinked lines. When animals were anesthetized, blood was drawn via the saphenous vein.

### 2.5. Blood Gases and Prothrombin Time Test

Approximately 150 µL of untreated, uncoagulated whole blood filled the sample ports of a CG4+ blood gas/lactate/pH analysis cartridge and a PT/INR prothrombin time test cartridge in an i-STAT handheld analyzer (Abbott Laboratories, Abbott Park, IL, USA), as previously described [42].

### 2.6. Blood Chemistry

The serum supernatant from each time point was collected and tested with a Piccolo General Chemistry 13 reagent disc on a Piccolo point-of-care blood chemistry analyzer (Abaxis, Union City, CA, USA), as previously described [42]. Serum was evaluated for albumin, alkaline phosphatase (ALP), alanine aminotransferase (ALT), amylase, aspartate aminotransferase (AST), blood urea nitrogen (BUN), calcium, creatinine (CRE), gamma-glutamyltransferase (GGT), glucose, total bilirubin, total protein, and uric acid. The remaining serum samples were aliquoted and stored at −80 °C until use in plaque assays or qRT-PCR. 

### 2.7. Cytokine and Chemokine Analysis

Circulating cytokines and chemokines were analyzed at each time point of the study using a Milli-Plex MAP NHP pre-mixed 23-plex assay (Millipore). Whole blood was directly collected from the catheter and aliquoted into K2EDTA MiniCollect blood tubes (Greiner), which were centrifuged at 3000× *g* for 10 min. The plasma supernatant was then aliquoted and stored at −80 ± 10 °C. Frozen tubes were allowed to thaw at room temperature prior to use in the cytokine assay. Samples were centrifuged at 5000× *g* for 10 min in order to remove any cryoprecipitate formed during storage. In accordance with the manufacturer’s instructions, washings were performed with a Bio-Rad II wash system. The Bio-Plex 100 xMAP system (Bio-Rad), in conjunction with BioPlex Manager 5.0 software, was used to acquire data in accordance with the manufacturer’s instructions.

### 2.8. D-Dimers

Blood derived from the CVC was aliquoted into 1 mL sodium citrate MiniCollect tubes (Greiner). The citrated blood sample was centrifuged at 3000× *g* for 30 min, and then the plasma supernatant was transferred into a fresh tube and stored at −80 ± 10 °C. The D-dimer quantification of samples was determined using an Asserachrom D-Dimer Enzyme Immunoassay kit (Diagnostica Stago Inc., Parsippany, NJ, USA) based on the manufacturer’s procedures. The citrate plasma was allowed to thaw at room temperature and then centrifuged at 3000× *g* for 15 min to remove any cryoprecipitate. The lyophilized standard was prepared during centrifugation to yield a fibrinogen equivalent unit (FEU) concentration between 950 and 1100 ng/mL; the actual D-dimer concentration of the standard was half of the FEU. The standard curve was prepared by diluting the 1:21 stock provided with an assay diluent buffer followed by two-fold serial dilutions to 1:42 and 1:252 and six-fold serial dilutions to 1:1512 and 1:9072, thus ensuring that data points fell within the linear range. All other aspects of the assay were performed in accordance with the manufacturer’s instructions. Plate washes were performed using Bio-Rad Pro Wash II using the standard plate adapter. Data were acquired via SoftMax Pro software (version 5.3) on a SpectraMax M5 (Molecular Devices, Sunnyvale, CA, USA) plate reader at an absorbance of 450 nm.

### 2.9. Flow Cytometry

CD45 PERCP (Clone D058-1283), CD3 V450 (Clone SP34-2), CD20 PE (Clone L27), and CD14 FITC (Clone M5E2) antibodies were purchased and used according to the manufacturer’s instructions (BD Biosciences). Fluorochrome-labeled monoclonal antibody cocktails were prepared with a BD FACS Stain Buffer in a 96-well plate. Whole blood was added to each well containing the antibody cocktail, and plates were vortexed and then incubated in the dark for 10 min. A BD FACS Lyse solution was used to fix leukocytes and lyse red blood cells in the sample for an additional 15 min prior to centrifugation at room temperature at 450× *g* for five minutes. Cell pellets were washed in a BD FACS Stain buffer followed by an additional centrifugation step prior to resuspending and analyzing cells on an LSR II Fortessa (BD Biosciences). Staining controls and compensation for each stain were prepared according to the manufacturer’s instructions (BD Biosciences). 

Compensation controls were run each day samples were analyzed. Compensated events were exported from BD FACS Diva (Becton–Dickinson). Data were imported into Cytobank (Cytobank Incorporated) and gated to select granulocytes (SSC-Ahi FSC-Ahi), monocytes (CD3–CD20–CD45 and SSC-Alo-mid FSC-Amid), T cells (CD3, CD45, and SSC-Alo-mid FSC-Amid), and B cells (CD20, CD45, and SSC-Alo-mid FSC-Amid). CD14 staining was absent, so monocytes were defined as CD3–CD20–. Due to altered scattering parameters at late time points, monocytes and lymphocytes were gated together on scattering parameters and separated only by CD3/CD20 gates. Counting beads were not run with samples, so absolute cell counts could not be determined. Instead, we report the percent of each cell type out of all non-debris events (granulocyte SSC-Ahi FSC-Ahi events plus lymphocyte/monocyte SSC-Alo-mid FSC-Amid events).

### 2.10. Hematology

The analytic performance of the Hemavet 950F hematology analyzer (Drew Scientific Inc., Dallas, TX, USA) was assessed using a series of stabilized controls (high, normal, and low) provided by the manufacture prior to analyzing blood samples. The analyzer was recalibrated using values obtained from the complete blood count (CBC) controls when control values were not within the manufacturer’s acceptable range. Once parameters for the instrument were verified to be within the manufacturer’s standards, blood samples were analyzed daily from each NHP between times of 0730 and 1200. Blood samples in 0.5 mL K2EDTA MAP tubes were inverted 8–10 times and probed with a Hemavet 950FS hematology analyzer (Drew Scientific Inc.) to obtain the CBC profile. 

### 2.11. Plaque Assays

Virus titration was performed via a plaque assay using Vero E6 cells (ATCC, Manassas, VA, USA). This cell line was grown in 10% fetal bovine serum (FBS) supplemented with α-MEM (Gibco). Vero E6 cells were plated in sterile 6-well plates and used in experiments once they reached greater than 90% confluence. Media was then removed from each well and replaced with 200 µL of dilution or material to be tittered. Viral inoculums were incubated on the cell monolayer for one hour in a humidified incubator at 37 °C and 5% CO_2_.

After the incubation, cellular monolayers were covered with the microcrystalline methylcellulose semi-solid Avicel-591 (FMC Biopolymer, Philadelphia, PA, USA) for a 1.25% final concentration in a growth medium. These cells were covered for 8–9 days in a humidified incubator at 37 °C with 5% CO_2_. After this period, 0.4% crystal violet in a neutral buffered formalin solution was added to each well and incubated overnight. All wells were washed with water, and plaques were counted. Titers were calculated by multiplying the dilution factor for each condition by the averaged plaque counts. The calculated titers were expressed as plaque-forming units per milliliter (PFU/mL).

### 2.12. qRT-PCR

At the time of necropsy, tissues harvested were processed via cutting, weighing, and homogenizing to produce 10% *w/v* tissue homogenates in α-MEM-containing Glutamax and 10% FBS (Gibco). Tissue RNA isolation and qRT-PCR reactions were prepared as previously described [23]. The PCR amplification of the EBOV glycoprotein gene was performed using a forward primer (1 µM (TTT TCA ATC CTC AAC CgT AAg gC), a reverse primer (1 µM (CAg TCC ggT CCC AgA ATg Tg), and a TaqMan probe (0.1 µM (CAT gTg CCg CCC CAT CgC TgC)) at 50 °C for 15 min. Initial denaturation was 95 °C for 5 min; this was followed by 45 cycles of denaturation at 95 °C for 1 s; annealing, synthesis, and single acquisition at 60 °C for 20 s; and final cooling at 40 °C for 30 s. A viral standard curve was developed through the ten-fold serial dilution of the stock virus in the challenge matrix and later RNA extraction. The LightCycler 480 software (version 1.50) was then used to quantify viral RNA based on a viral RNA standard.

## 3. Results

### 3.1. Clinical Observations

Four healthy rhesus macaques were exposed to a high target dose of approximately 1000 PFU of EBOV (Kikwit strain) via aerosol exposure, as described in Appendix A. Henceforth, mean clinical score and observations were used when determining stages of EVD progression. No observable changes were noted from DPE 0 to 4, which was defined as the subclinical stage (Appendix A). The initial clinical observation on day post-exposure (DPE) 5 occurred with the lack of appetite in NHP4. All NHPs experienced a lack of appetite on DPE 6, at which NHP2 and NHP3 did not pass any stool.

Alterations in the responsiveness of all animals were observed between DPE 6 and 8. Changes in posture and appearance occurred within DPE 6–10 (Appendix A). All NHPs also had facial rash from DPE 6 to 10, except for NHP2. NHP4 had facial rash localized around the eyes on DPE 6, whereas NHP1 and NHP3 presented rashes on DPE 8. These rashes covered less than 50% of the body surface for NHP4 and NHP1, as well as more than 50% for NHP3. A mean clinical score of ≥2.0 was observed after DPE 6, thus marking the decompensated stage (Appendix A). NHP2 and NHP4 succumbed on DPE 7, with minor clinical observations taken throughout disease progression (Appendix A). NHP1 met euthanasia criteria on DPE 8 based on clinical score equaling 10. NHP3 was euthanized on DPE 10 because the animal had a clinical score of 8 with indications of having a seizure. 

### 3.2. Viremia

EBOV serum titers were determined by standard plaque assays described in Section 2.11. Serum titer levels were initially detected in NHP4 on DPE 5, NHP3 on DPE 7, and NHP2 and NHP1 on DPE 6 (Figure 1a). Peak titer levels occurred by DPE 7. NHP3 had lowest peak titer levels on DPE 7 (24 × 10^3^ PFU/mL) compared to other challenged animals. NHP3 survived longer than other NHPs experiencing high viremia.

EBOV RNA was analyzed using the qRT-PCR procedures described in Section 2.12, and it was detected in multiple tissues at terminal endpoints from the NHP (Figure 1b). The highest amount of EBOV RNA equivalents in descending order were the spleen, liver, mesenteric lymph nodes, gonads, axillary lymph nodes, adrenal gland, tracheobronchial lymph nodes (only collected in NHP1), inguinal lymph nodes, bone marrow, heart, mandibular lymph nodes, lung, kidney, pancreas, and lastly the brain. Based on the sum in all tissues, NHP4 had the greatest detectable levels (6.8 × 10^7^ PFU/mL), followed by NHP1 (3.8 × 10^7^ PFU/mL), NHP2 (2.4 × 10^7^ PFU/mL), and NHP3 (1.3 × 10^6^ PFU/mL). This EBOV RNA abundance reflects viral tissue levels at the time of necropsy and is not reflective of viral tissue levels at different stages of disease.

### 3.3. Telemetry

Radio telemetry devices were implanted into all NHPs to survey continuous real-time physiologic parameters. Diurnal variations were observed in baseline telemetric readings of temperature, respiratory rate, mean blood pressure, and heart rate for all NHPs (Figure 2 and Appendix A). Normal diurnal variation was observed, with each parameter rising in the mornings and falling in the evenings. All NHPs had a complete loss of diurnal variation for temperature, respiratory rate, mean blood pressure, and heart rate by DPE 5 compared to baseline readings (Figure 2 and Appendix A). 

Fever is the primary indicator of the clinical stage. It is defined as a temperature greater than 1.5 °C above baseline for longer than 2 h, which was observed in all animals on DPE 5 accompanied by a loss of diurnal variation (Figure 2). The mean peak temperature of 40.73 ± 0.19 °C occurred on DPE 5–6. Hyperpyrexia, marked by a temperature greater than 3 °C above baseline for longer than 2 h, was observed in all NHPs between DPE 5 and 7. On the day of death, all animals experienced a drop in body temperature. 

All animals developed and sustained elevated respiratory rates, with values greater than 3 SD above baseline on DPE 5, until entering a moribund state (Appendix A). The peak respiratory rate of 50 ± 5 breaths/minute nearly doubled the DPE 0 value of 27 ± 2 breaths/minute. Mean blood pressure was increased by greater than 3.0 SD on DPE 5, but overall readings resembled baseline (Appendix A). These readings were substantially less than 3.0 SD prior to death, in which the lowest mean value for animals was 52 ± 9 mmHg in contrast to a baseline of 111 ± 7 mmHg on DPE 0. No mean blood pressure values for NHP2 were detected due to a sensor malfunction.

Relative to the clinical stage, the heart rate of NHPs was significantly higher than 3 SD from corresponding baseline on DPE 5–6 (Appendix A). Electrocardiograms identified changes in diurnal variations for PR, QTc Fridericia, RR, and QRS durations on DPE 5 (Appendix A). The mean peak heart rate for EBOV-challenged animals was 211 ± 7 beats/min by DPE 7 versus 141 ± 8 beats/min on DPE 0, and then the heart rate decreased in all the animals when entering a moribund state. The percentage of mean PR (14%: Appendix A), QT-corrected Fridericia (13%: Appendix A), and RR (27%: Appendix A) durations decreased on DPE 7 compared to baseline. These telemetric readings were inversely correlated to the heart rate values. 

All animals presented high QRS duration values by DPE 7 (6.8%: Appendix A), except for NHP4. In addition, three of four NHPs developed ST segment morphology changes, two NHPs had QRS complex morphology changes, and one NHP had T-wave inversions (Appendix A). NHP4 had the shortest QRS duration of the NHPs (Appendix A). Changes in QRS morphology and the related S-point detection in NHP4 (Appendix A) corresponded with a shorter QRS duration (Appendix A). For all other subjects, a terminal QRS duration increase was associated with elevated heart rate (Appendix A). The clinical stage of disease was accordingly found to include elevated body temperature, respiratory rate, heart rate, and mean blood pressure with changes in diurnal variation (DPE 5–6). The decompensated stage was marked by decreased body temperature and mean blood pressure (after DPE 6).

### 3.4. Blood Gases and Clinical Chemistry

Changes in venous blood gas parameters were observed during EVD progression beginning on DPE 2 (Figure 3). In comparison to DPE 0, the group mean of venous partial pressure of carbon dioxide (pCO_2_: Figure 3a) increased by 10.8% on DPE 2 and decreased on DPE 3. Venous blood gas levels on DPE 4 decreased by 11.3% for pCO_2_, 20.3% for total carbon dioxide (tCO_2_), 18.9% for bicarbonate, and 136.8% for base excess of extracellular fluid (BEecf) compared to the baseline levels (Figure 3a–d). All animals had low pH on DPE 2 (7.43 ± 0.04), particularly NHP2 (7.40) and NHP3 (7.39), in comparison to baseline pH levels (7.48 ± 0.01: Figure 3e). These animals had high pH levels (7.58 ± 0.05) on DPE 6, which decreased below baseline prior to death (7.37 ± 0.10). All animals had increased lactate concentration starting DPE 7 (Figure 3f).

The venous partial pressure of oxygen (19.3%; Appendix A) and soluble oxygen percentages (11.9%; Appendix A) increased on DPE 2 compared to baseline. Beginning on DPE 3, venous pO_2_ and sO_2_% levels fluctuated until near death, when levels increased. These findings are suggestive of respiratory acidemia during the subclinical stage (DPE 2), respiratory alkalosis over the clinical stage (DPE 5–6), and metabolic acidemia toward the end of the study (DPE 7–10) associated with increased lactate concentrations. 

Serum ALT (978.75 ± 724.08 IU/L; Figure 4a), AST (1631.50 ± 265.72 IU/L; Figure 4b), GGT (244.00 ± 20.64 IU/L; Figure 4c), and ALKP (506.25 ± 130.89 IU/L; Figure 4d) activities increased on day of death (DPE 7–10) compared to baseline. Animals also experienced increased serum total bilirubin (1.23 ± 0.43 mg/dL: Appendix A), blood-urea-nitrogen (87.50 ± 63.08 mg/dL: Appendix A), and creatinine (4.73 ± 2.57 mg/dL: Appendix A) concentrations on the day of death in comparison to baseline [45].

Glucose concentrations notably increased on DPE 4 (109.25 ± 26.32 mg/dL; Appendix A), and then mean endpoint concentrations (35.25 ± 19.00 mg/dL) dropped below baseline by 51%. Trends of serum calcium, amylase, total protein, and albumin decreased throughout the time course relative to baseline, except for the fluctuation observed with NHP4 on DPE 2 and NHP2 on DPE 6 (Appendix A). The mean endpoint concentrations of serum calcium (30%; Appendix A) and amylase (46%; Appendix A) decreased below baseline. The mean endpoint concentrations of total protein (12%; Appendix A) and albumin (32%; Appendix A) also decreased below baseline. 

### 3.5. Hematology

All EBOV-challenged animals experienced increases in hematocrit on DPE 6 (20%; Figure 5a), which included numbers of circulating red blood cells (18%; Figure 5b), when compared to DPE 0. Hemoglobin concentrations also increased on DPE 6 (17%; Figure 5c). These numbers and concentrations returned to near baseline at endpoints of animals. Mean corpuscular volume, mean corpuscular hemoglobin, mean corpuscular hemoglobin concentrations, and red blood cell distribution width percentages displayed wide ranges near baseline throughout the time course (Appendix A). 

The numbers of white blood cells fluctuated throughout the time course near baseline, except for high numbers observed in NHP2 and NHP3 on DPE 6 (Figure 5d). Reductions of circulating lymphocyte numbers were substantially evident on DPE 4 (43%; Figure 5e) and continued below baseline levels after DPE 7. The number of circulating monocytes fluctuated near baseline levels on DPE 0, and then increased at the terminal endpoint compared to the previous day (Figure 5f). High numbers of monocytes were distinctly observed for NHP1 on DPE 2 and NHP3 on DPE 10. 

Neutrophil numbers increased on DPE 5 (64%; Figure 5g) and then decreased near terminal endpoints with the exception of NHP1. Similar to the white blood cells, the number of neutrophils in NHP1 closely resembled baseline levels throughout the time course. Basophil numbers increased on DPE 6 (77%; Figure 5h) when compared to baseline levels and continued to increase for each NHP until the terminal endpoint. With the exception of NHP3, the number of eosinophils decreased in all NHPs on DPE 6 (118%; Figure 5i) compared to baseline. Similar to monocytes, high numbers of eosinophils were observed solely in NHP1 on DPE 2 and NHP3 on DPE 10. 

### 3.6. Immune Cells and Cytokine Levels in Blood of Aerosol EBOV-Challenged Rhesus Macaques

The flow cytometric analysis of granulocyte, monocyte, and T and B lymphocyte percentages was conducted with the blood of EBOV-challenged NHPs. The percentage of granulocytes was elevated on DPE 6 (102%; Appendix A) in comparison to baseline levels. Consisting of predominately neutrophils, this increase in the percentage of granulocytes decreased at the terminal endpoint. The trends in granulocyte percentages and neutrophil numbers (Figure 5g) for each NHP were similar. However, NHP1 had elevated granulocyte percentages on DPE 5, unlike neutrophil numbers, when compared to DPE 0. 

Monocyte percentages decreased on DPE 5 (79%; Appendix A). Similar to the results shown by the hematological assessment of monocytes (Figure 5f), the percentage of monocytes then mildly increased after DPE 5 until reaching the terminal endpoint of each NHP. Complimentary to lymphocyte number (Figure 5e), CD3^+^ T cell percentages were below baseline levels on DPE 5 and lowest on DPE 6 (65%; Appendix A). Similar to lymphocyte numbers, this increase in the percentage of granulocytes increased at the terminal endpoint of each NHP. The percentage of CD20^+^ B cells was also below baseline levels on DPE 6 (62%; Appendix A). Trends for the rise, decline, and rise of CD20^+^ B cell percentages were observed in all animals during disease progression, with delay observed in initial rise for NHP4. Comparably, the time course of lymphocytes consisted of a mild increase in cell numbers on DPE 2, a decrease between DPE 4 and 5, and lastly a mild increase near the terminal endpoint.

Cytokine analysis identified the initial blood levels of IFNγ, IL-18, MCP-1, IL-1RA, IL-1β, IL-2, IL-5, IL-6, IL-8, IL-12p40, MIP-1β, TNFα, GM-CSF, and G-CSF on DPE 6 (Figure 6a–c and Appendix A). The peak level of cytokines occurred after DPE 6 during the decompensated stage of EVD, with the exception of the earlier responses for IL-1β in NHP2 and IL-6 in NHP4. Relative to a moderate viral presence, NHP3 had a delayed and attenuated upregulation of cytokines in comparison to other NHPs. 

### 3.7. Blood Coagulation

Platelet numbers decreased in all NHPs on DPE 6 (37%; Figure 7a) and DPE 7 (69%), suggestive of thrombocytopenia [45]. The mean platelet volumes decreased on DPE 3 (17%; Figure 7b), returned to volumes close to DPE 0, and then mildly decreased (19%) at each terminal endpoint. Plasma D-dimer concentration, a known marker for disseminated intravascular coagulation, increased in NHP 3 on DPE 4, NHP4 on DPE 7, and NHP1 and NHP2 on DPE 6. High levels of plasma D-dimers were observed in the terminal endpoints of all NHPs (Figure 7c). Extensions of prothrombin times were evident in all animals on DPE 7 (37%; Figure 7d). To substantiate prothrombin times, the international normalized ratio (INR) was calculated and was also found to have increased on DPE 7 (41%; Figure 7e). 

## 4. Discussion

Ebolavirus is a known filamentous mammalian virus reported to infect constitutively active macropinocytotic cells, such as dendritic cells and macrophages, and extravasate into bodily tissues [1,2,3,6,11,35,46,47,48,49]. This study followed the clinical course of aerosol-induced disease of four rhesus macaques along with pathological assessments published by Twenhafel et al. in 2012 to capture time-dependent changes during EVD progression. In comparison to historic intramuscular studies, the in-depth analyses of aerosol studies support countermeasure development such as vaccines and therapeutics.

A detailed evaluation for the time course of acid—base status was studied using the telemetric detection of fever—the primary marker for the clinical stage of aerosol-induced EVD. During the subclinical stage, venous levels of partial pressure of carbon dioxide were found to be inversely related to low pH levels suggestive of respiratory acidemia in three out of four NHPs. There is minimal published evidence for this finding, and the etiology is not well-understood. A reason for this change was not identified by alterations in respiratory rate, errors in the administration of virus, instrument and sampling issues, or any faults in study design. Further research is required to identify whether viral expansion influences local cells, lymphocytes, and additional blood components to decrease pH levels.

The venous levels of partial pressure of carbon dioxide, total carbon dioxide, bicarbonate, and base excess of the extracellular fluid decreased in NHPs late in the subclinical stage. In comparison to this aerosol study, intramuscular studies by Warren et al. in 2020 identified a similar trend for total carbon dioxide levels from fever until death [23]. The difference in timing for total carbon dioxide depression in aerosol and intramuscular routes could have been due to waning dietary consumption, hydrogen ion loss from vomiting, and loss of carbon dioxide relative to changes in respiratory rate for thermoregulation. All rhesus macaques experienced respiratory alkalosis late in the clinical stage. Subsequently, cardiac conduction abnormalities were observed for all animals in the decompensate stage with differences relative to viremia and EBOV levels in the heart. Similar to intramuscular findings, disease progression was ensued by terminal decline in venous total carbon dioxide, hypotension, elevated lactate concentrations, metabolic acidemia, shock, and death in the decompensated stage as observed in Figure 8 [23,37].

The characteristics of intramuscular and aerosol-induced EVD using target dose of 100 PFU in NHPs were reportedly nearly identical when comparing peak viremia, infectious virus profile, survival, and time to death [50]. The present aerosol-induced EVD findings identified an extended subclinical stage by one day and shortened clinical stage with no change in terminal endpoint. No major differences were observed at the start and endpoints of the decompensated stage.

EBOV glycoprotein and nucleic acid were identified within the lungs and respiratory lymph tissues early in the related aerosol serial sacrifice study [35], where the virus spread from the lungs and associated lymph tissues into the spleen, liver, and heart. These tissues consisted of more than 10^3^ PFU/mL near the of subclinical stage. The infected blood monocytes and free virus disseminated into accessible target tissues were accompanied by fever. The large surface area of the respiratory tract and low abundance of constitutively active macropinocytotic cells are believed to affect the low viral load in the lungs and the delayed/shorten duration of the clinical stage of aerosol-induced EVD. In this study, endpoint viral titers of the lungs were lower than known targets in the following descending order: spleen, liver, mesenteric lymph nodes, gonads, axillary lymph nodes, and adrenal glands.

The attrition of platelet and lymphocyte numbers has been shown in both intramuscular and intraperitoneal studies of NHPs [20,23,35,51]. These reductions in platelet numbers and mean platelet volume, inversely correlated to elevated prothrombin time and D-dimers, were followed by petechiae in all but one NHP. Uniquely, in vitro studies of infected primary human monocytes have identified the increased expression of extrinsic coagulation factors as early as one hour post-inoculation [20]. These findings suggest that platelets and the blood coagulation pathway are impacted by EBOV prior to the appearance of fever.

Apoptotic splenic lymphocytes have been identified prior to fever within the germinal centers and mantle zones [35]. EBOV is known to induce the abortive infection of T cells through ER stressed-induced autophagy and cell death [52]. EBOV also has been reported to block dendritic cell maturation, which prevents the activation of uninfected lymphocytes, thus resulting in apoptosis [22,35,53,54]. Our data support these findings with the attrition of blood lymphocyte numbers and percentages of CD3^+^ T cells during the subclinical stage. The percentage of B cells mainly decreased during the clinical stage, similar to the results of intramuscular-induced EVD model [38]. Though autoimmune responses are known late in infection, we hypothesize that B cells contribute to anti-glycoprotein antibodies and autoimmune responses involving anti-DNA and anti-heat shock protein 60 antibodies that influence pathogenesis [38,55,56,57]. 

Monocytes/macrophages and dendritic cells, parts of the mononuclear phagocyte system, are permissive to EBOV infection and replication without inducing significant apoptosis [20,35]. EBOV multifunctional protein (VP35) is reported to participate in viral RNA synthesis, inhibit interferon regulatory factor-3 (IRF3), and block antiviral responses [58,59,60]. Accordingly, filovirus-infected macrophages and dendritic cells are reported to appear normal or necrotic [20]. The declines in the percentages of monocytes and viremia in the current work indirectly support these highlighted studies.

Our hematological assessment identified increases in white blood cell, neutrophil, and basophil numbers during the clinical stage, similar to previous intramuscular and aerosol studies [23,30,37]. The activation of neutrophils has a role in the release of pro-inflammatory mediators and tissue damage during EVD [56,61,62]. Viral replication was uniquely not detected in neutrophils [61]. Hematocrit, red blood cells, and hemoglobin levels were also elevated, indicative of hemoconcentration due to dehydration. Distinctly, the reduction of lymphocyte and monocyte numbers are followed by decreased eosinophils counts. These decreases in the number of eosinophils and related antiviral responses through release of RNases, DNAses, and interferons, as well as interactions with infected monocytes, require further investigation [63,64]. 

Blood cytokine and chemokine levels begin to increase in parallel with hematocrit during the decompensated stage. This cytokine storm contributes to liver, kidney, and gastrointestinal inflammation through a bystander effect [23,35] whereupon NHPs succumb to EVD through viremia, the depletion of nutrients, and life-threatening changes in vital signs. The early implementation of fluid and nutrient replenishment can attenuate the detrimental effects of the disease in intramuscularly infected NHPs [65]. The benefits of such advanced supportive care in combination with cutting-edge antiviral therapeutics require investigation to discover the impact on disease progression.

## 5. Conclusions

This natural history study of aerosol-induced EVD provides detailed telemetric, clinical, laboratory, and immunological parameters. This study has highlighted potential early disease markers, types of acid–base disorders, and differences in stages of disease for aerosol and intramuscular models. The timeframe for the clinical detection of the aerosol model was shortened compared to an intramuscular model. Therefore, the identification of disease stages could indicate a time point for implementing biodefense countermeasures.

## Figures and Tables

**Figure 1 viruses-13-02297-f001:**
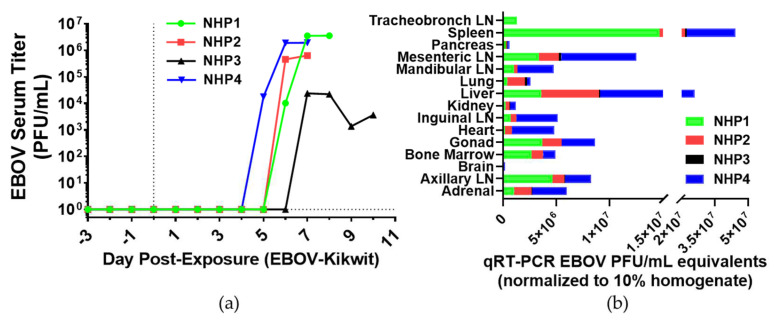
Viral detection in sera and tissues of EBOV-challenged rhesus macaques. (**a**) Plaque titer (PFU/mL) of serum for each NHP at specific time points. DPE 0 is marked by a black vertical dot line. The black horizontal dashed line is the average of all animals for DPE 0. (**b**) EBOV RNA equivalent levels at terminal endpoint of NHPs for respective tissues; lymph node (LN). RNA was amplified using qRT-PCR to identify Ebola virus glycoprotein gene in tissue of each NHP. NHP color designation is the same as (**a**).

**Figure 2 viruses-13-02297-f002:**
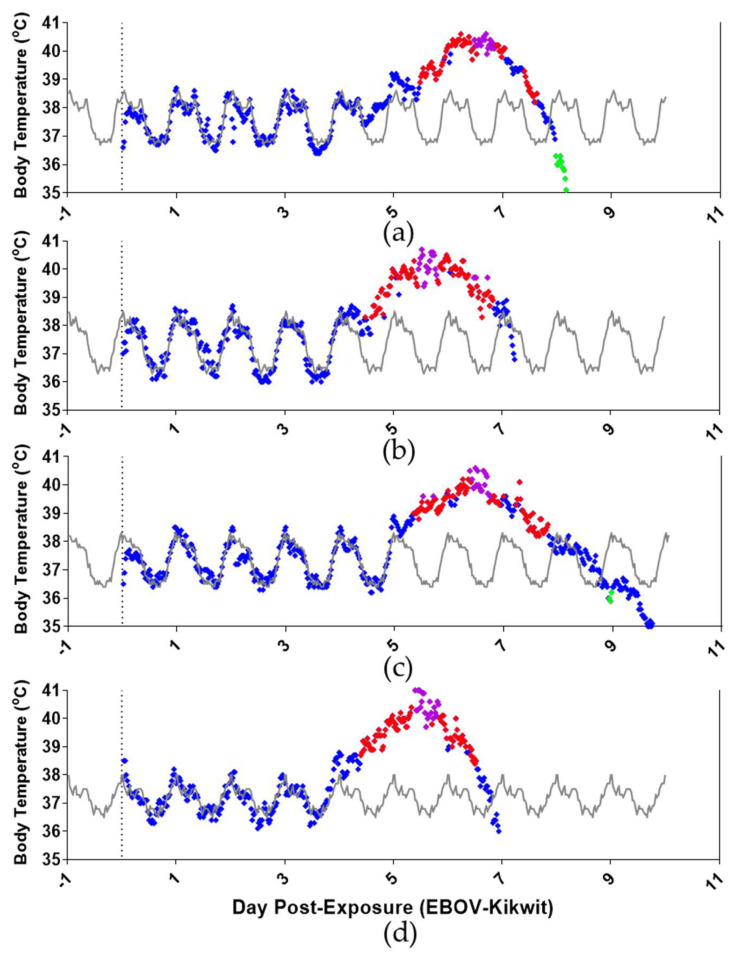
Telemetric temperature detection of aerosol EBOV-challenged rhesus macaques: (**a**) NHP1, (**b**) NHP2, (**c**) NHP3, and (**d**) NHP4. Vertical grey dotted line marks DPE 0—BL (baseline), ♦ marks fever (>1.5 °C over baseline) for longer than 2 h, ♦ marks LS (values significantly lower: <3.0 SD from corresponding baseline) for longer than 2 h, ♦ marks temperature (°C) for longer than 2 h, and ♦ marks hyperpyrexia (>3.0 °C over baseline) for longer than 2 h.

**Figure 3 viruses-13-02297-f003:**
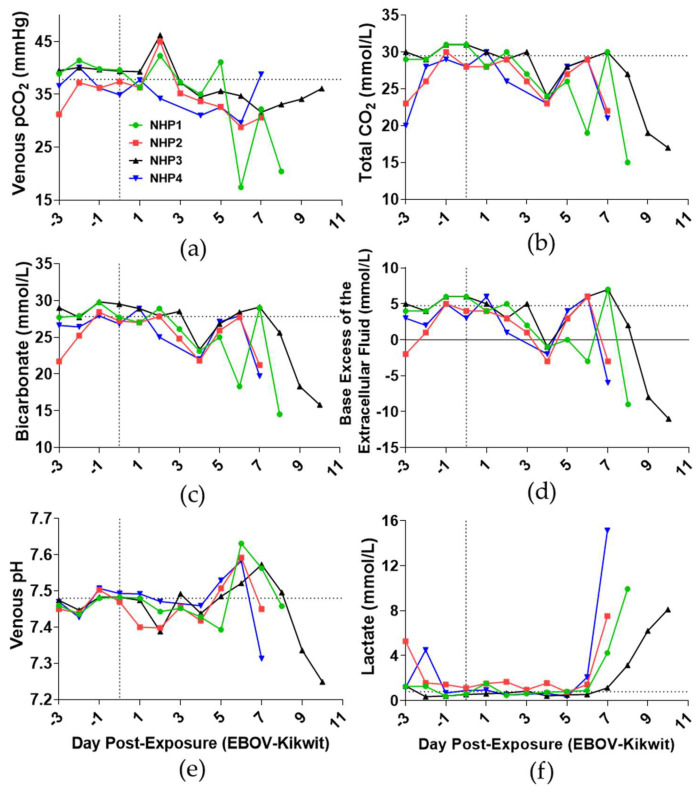
Time course of blood gases and constituents of EBOV-challenged rhesus macaques. Detection of (**a**) partial pressure of carbon dioxide (pCO_2_), (**b**) total CO_2_, (**c**) bicarbonate, (**d**) base excess of extracellular fluid, (**e**) pH, and (**f**) lactate were determined in whole blood using i-Stat CG4+. DPE 0 is marked by a grey vertical dot line. The horizontal dashed line is the average of all animals for DPE 0.

**Figure 4 viruses-13-02297-f004:**
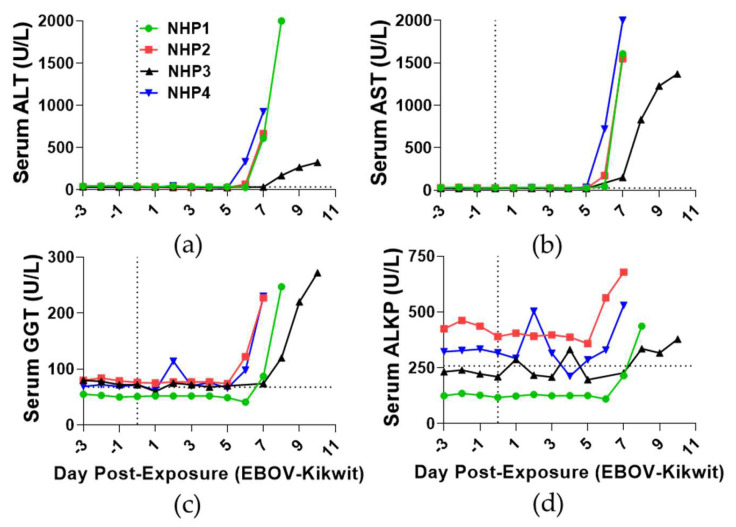
Time course of serum liver enzyme activity of EBOV-challenged rhesus macaques. Serum levels of (**a**) alanine aminotransferase, (**b**) aspartate aminotransferase, (**c**) gamma-glutamyl transferase, and (**d**) alkaline phosphatase were detected using Piccolo chemistry. DPE 0 is marked by a black vertical dot line. The black horizontal dashed line is the average of all animals for DPE 0.

**Figure 5 viruses-13-02297-f005:**
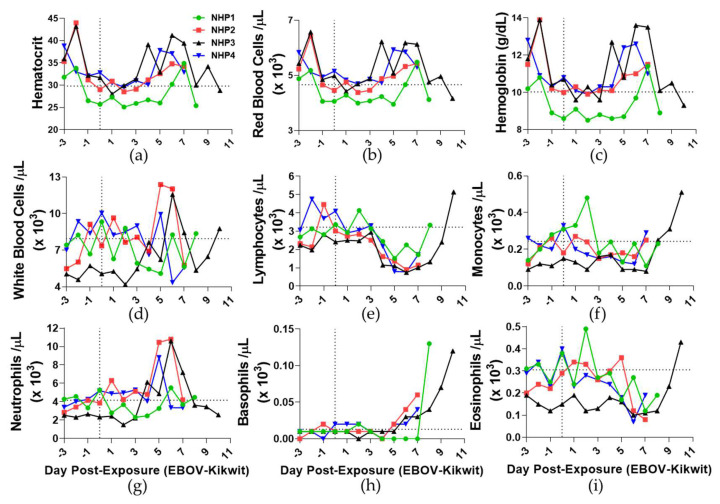
Time course analysis of red blood cell mass and white blood cells from aerosol EBOV-challenged cynomolgus rhesus macaques: (**a**) hematocrit percentage, (**b**) number of red blood cells, (**c**) hemoglobin concentration, (**d**) number of white blood cells, (**e**) number of lymphocytes, (**f**) monocytes, (**g**) number of neutrophils, (**h**) number of basophils, and (**i**) number of eosinophils detected per microliter of blood. DPE 0 is marked by a black vertical dot line. Black horizontal dashed line is average of all animals for DPE 0.

**Figure 6 viruses-13-02297-f006:**
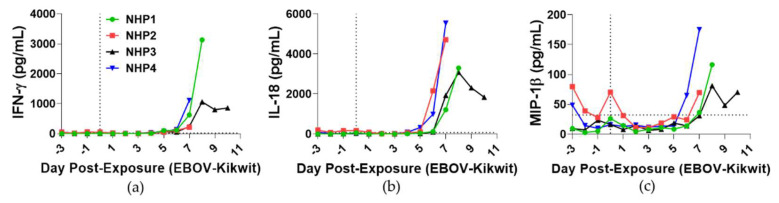
Analysis of immune responses in blood from aerosol EBOV-challenged rhesus macaques. Cytokine analyses of blood—(**a**) IFNγ, (**b**) IL-18, and (**c**) MCP-1—were performed on samples collected 3 days prior to exposure and DPE 0–10. DPE 0 is marked by a black vertical dot line. Black horizontal dashed line is the average of all animals for DPE 0.

**Figure 7 viruses-13-02297-f007:**
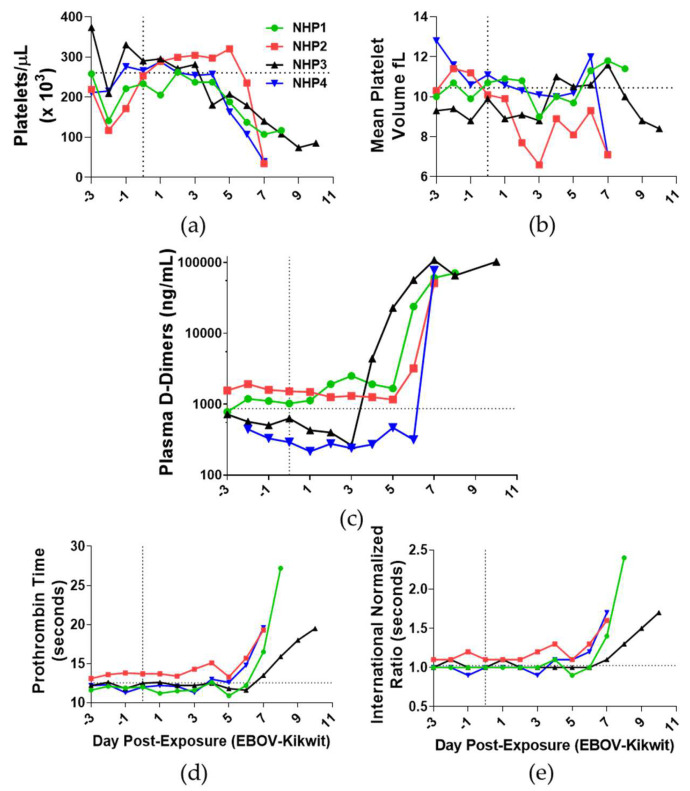
Assessment of platelets and clot formation in the blood of aerosol EBOV-challenged rhesus macaques. Blood (**a**) platelets, (**b**) mean platelet volume, (**c**) plasma D-dimers, (**d**) prothrombin time, and (**e**) international normalized ratio were analyzed in samples collected 3 days prior to exposure and DPE 0–10. DPE 0 is marked by a black vertical dot line. The black horizontal dashed line is the average of all animals for DPE 0.

**Figure 8 viruses-13-02297-f008:**
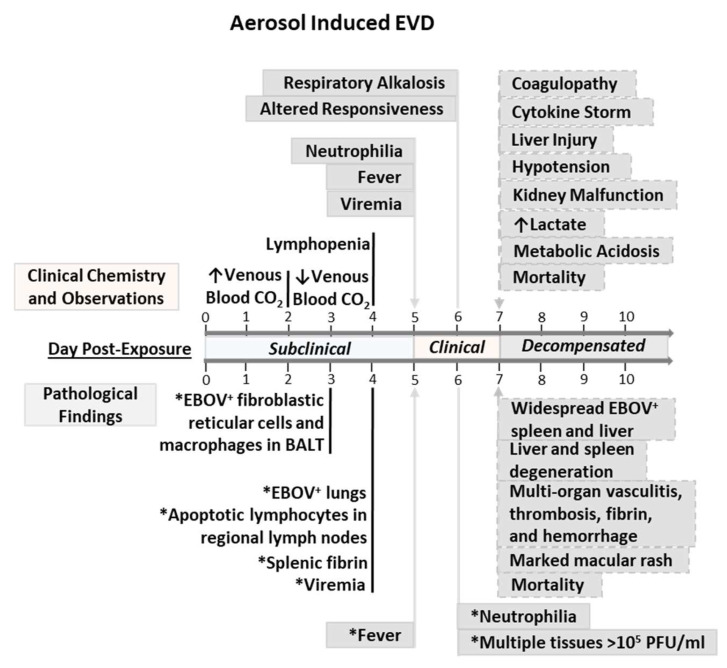
Natural history of aerosol-induced EVD in rhesus macaques. The subclinical stage of disease progression is defined by absence of fever between DPE 0 and 4. The clinical stage of disease progression is defined by fever, as observed on DPE 5 in the current study and the serial sacrifice study by Twenhafel et al. 2012. Multiple tissues (qRT-PCR EBOV-positive) are in descending order: inguinal LN, liver, spleen, tracheobronchial LN, lung, axillary LN, mandibular LN, kidney, adrenal gland, mesenteric LN, and bone marrow. The decompensated stage of the natural history study was identified by coagulopathy, cytokine storm, liver injury, kidney malfunction, and mean clinical score ≥ 2.0. Macular rashes were found on the face, pinna, arms, and legs of animals. Here, the aerosol-induced EVD had an extended subclinical stage, shortened clinical stage, and general decompensated stage when compared to historic intramuscular studies noted in Appendix A (lymph node = LN; solid black line = subclinical stage; solid gray arrow = start of clinical stage; solid gray line = clinical stage; and dashed gray arrow = start of decompensated stage). * Asterisk pertains to Twenhafel et al. 2012.

## Data Availability

Data is contained within the article or Appendix A. The data presented in this study are available in at https://www.mdpi.com/article/10.3390/v13112297/s1.

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
