# Peer review of "Natural History of Aerosol-Induced Ebola Virus Disease in Rhesus Macaques"

_viruses, 2021, doi:10.3390/v13112297_

Round 1

Reviewer 1 Report

Downs and coauthors describe a 4-animal descriptive infection study using aerosol modality of exposure and ebola virus.  The study is identified in the title as a 'natural history' but it is none of the sort; as this represents one cohort of a much bigger study in order to be defined as a natural history study. Generally NH studies either explore dose as it relates to exposure modality and ID agent, or provision is made for killing of small cohorts during the timecourse of disease for exploration of ongoing pathologic consequence temporally.  The current study targets one dose (1000 PFU inhaled) amongst all subjects, and 2/4 animals are euthanized whereas the others apparently died in cage.  It appears that the experimental design of this study is an afterthought, and not focused to truly determine the NH of aerosol disease.  This study should be framed/presented with other data, or possibly be augmented by accesory historical data for the purposes of publication.  A presentation of 4 animal infections - whose outcomes are disparate in certain endpoints - is niether a NH study nor a stand alone publication.  Also, at least a statement should be made on why there is not one monicum of pathology included in this paper.  There are pathologists listed in the authorship.

Some comments,

line 90-96 these are well-known methods for determining individualized inhaled dosing.  The description appears written as only vaguely familar with the process of animal infection by aerosol.  And, suprisingly, no individual dosing information is supplied, not even in the supplemental data. What was the variability amongst the 4 animals? Could this have been the reason that there was departure of some of the biological and physiological endpoints in NHP3?  None of this is resolved.

Defining distinct phases of disease 1)subclinical, 2) clinical, and 3) decompensatory, is not clear based upon the results from 4 animal infections.  It's puzzling how this can be the basis for defining distinct parts of temporal development of disease. 

line 437-438.  This is a repeat of the same information from the abstract and the introduction (morphology of virus/virion).

line 441-442 This is a repeat of justification of 'why' do this study similar to abstract and introduction.

line 449-453.  Puzzled by this statement.  honestly do not know what the authorship is attempting to say (that there was errors in the dosing?)

Figure 8 legend is inappropriate and entirely too long.  Figure 8 is confusing at best, and a mess at worst.  One can barely see the temperature trace over the words overlay. 

Table S2.  This is close to 'raw' catagorical data.  There is no summarization at all.  This is not helpful in any way to the reader to decipher as a supplement to this work.

Figure S1.  Why no explanation for NHP4 data all over the graph? divergent with the other animals.

Figure S2.  Where is the data for NHP4?

Figure S4. Why does NHP4 decrease at endpoint whereas NHP3 plateaus? Where is explanation?  Same for S5, S6, S7, S14

Figure S13 This figure needs careful explanation and interpretation.  None is given.

Author Response

Greetings Reviewer 1,

   The authors greatly appreciate your experience, time, and considerations for the submitted manuscript.  Kindly find the attached response to the comments. The revised manuscript and supplementary material will be sent to the "Viruses" point of contact. IF you have any questions, please feel free to contact me directly.

Sincerely, 

Isaac Downs

Reviewer 2 Report

In their manuscript entitled “Natural history of aerosol induced Ebola virus disease in rhesus macaques (Macaca mulatta),” Downs et al. provide a detailed description of disease and a thorough analysis of disease markers following exposure of macaques to aerosolized Ebola virus. A greater understanding of Ebola virus pathogenesis in key model organisms is critical for developing novel and effective countermeasures. Not only does this study describe pathogenesis following an aerosol exposure, but it also complements and extends our understanding of Ebola virus pathogenesis in general. This study is well suited for publication and will be of interest to many in the field. Nevertheless, a few minor comments, outlined below, should be addressed prior to publication.

  • The authors state in the Discussion section (Line 489) that this study expands upon a study published much earlier (Twenhafel et al. Vet Pathol 2013), in which the four animals described here were used as controls for a serial sacrifice study. It would be helpful if this fact were stated upfront, either in the Introduction section or, at the very least, the Materials and Methods section. There is some key information in the earlier paper that may provide important context for interpreting the results of the present paper.
  • Similar to the above point, it would be helpful if the authors used the same animal ID numbers in the present manuscript as they used for the earlier manuscript. This would aid in assimilation of the two data sets.
  • The authors should provide the name of the Ebola virus isolate using the recommended nomenclature (PMID: 23001720). Additionally, the authors should provide the Genbank accession number for the sequence of this isolate. It is not clear from the references provided in Section 2.1 of the Materials and Methods what these key pieces of information are.
  • On Lines 95-96, the authors state that the inhaled dose of virus was calculate by back titration. The authors should provide the back titration values for each animal. I recognize that these values are provided in the Twenhafel et al. paper, but it would be useful to have them here as well.
  • Section 2.12 of the Materials and Methods outlines the procedures for “serum RNA isolation and qRT-PCR reactions,” but it is not clear where these data appear in the manuscript. Serum titers calculated by plaque assay are provided in Figure 1a, and RNA levels in tissues are provided in Figure 1b, but serum RNA levels appear to be missing. Please clarify.
  • What are the units of RNA quantification in Figure 1b? The Materials and Methods section suggests they could be PFU equivalents, while the text (Lines 286-287) suggests they are copy numbers. Please clarify in both the Materials and Methods section and the Results section how RNA levels were quantified.
  • It can be difficult to summarize trends for each parameter in multiple different animals, but the authors should review their descriptions of all data to confirm that they are accurate. For example,
    • It does not appear accurate to state that monocyte counts decrease on DPE 5 and then increase near baseline levels on DPE 6 (Lines 391-392). From the graph in Fig. 5f, it appears as if there is little change in monocyte numbers on DPE 5, and almost now change on DPE 6 relative to the previous day (with the exception of NHP1). Numbers appear to reach baseline for two animals on DPE 7 and the other two on DPE 8.
    • The authors state that the number of eosinophils decreased in all NHPs on DPE 6 (Lines 394-395), but this does not appear to be the case for NHP 1.
    • With the exception of NHP3, none of the Plasma D-Dimer levels increase on DPE 4, as indicated on Line 431.
  • Line 24 and 83: “plague” should be corrected to “plaque”
  • Line 42: “Ebola viruses” should be one word (i.e., “Ebolaviruses”)
  • Line 47: It may be more accurate to state that “Ebola virus” rather than “Zaire ebolavirus” is a Category A biological agent. The former is the actual agent, while the latter is the species that incorporates the agent.
  • Lines: 285-287: Please specify from which tissue the indicated peak levels of RNA were found.
  • Line 307: Is it necessary to include “Hypothermia” in the figure legend since it does not seem to appear in the figure?
  • Line 341: Cite “Fig. 3” instead of “Fig. 4”.
  • Line 437-438: This sentence is redundant (cf. Lines 43-44) and should be deleted.

Author Response

Greetings Reviewer 2,

   The authors greatly appreciate your experience, time, and considerations for the submitted manuscript.  Kindly find the attached response to the comments. The revised manuscript and supplementary material will have to be submitted to MDPI point of contact.  Please contact me if you have any questions.

Sincerely, 

Isaac Downs
